# Guidelines for Formulating Anti-Pollution Products

**Niraj Mistry**

Independent Consultant, Mumbai 400093, India; niraj01mistry@gmail.com; Tel.: +91-99-8701-3431

**Abstract:** Anti-pollution skin care and cosmetic products are witnessing a significant growth in the last couple of years due to worsening air quality across the world, and increasing awareness and concern regarding the adverse impact of various environmental pollutants on skin. The various pollutants, like particulate matter, oxides of nitrogen and sulphur, ozone, and polyaromatic hydrocarbons damage skin by different mechanism resulting in skin dryness, loss of firmness, dark spots, uneven skin tone, fine lines and wrinkles, aggravation of acne, and inflammation. The task of developing globally harmonized products is very challenging due differences in skin types according to ethnicity, variation in seasonal weather pattern, differences in benefit expectations, and variances in personal care and cosmetics usage habits of consumers in different regions of the world. However, an increasing understanding about the mechanism by which various pollutants damage the skin manifesting into various extrinsic signs of skin damage and development of various actives that counter the impact of different environmental aggressors has helped formulators to develop different products and to establish efficacy by in vitro and in vivo tests. The article summarizes approaches for formulation development, and a list of few actives classified based in their mechanism action is given. A representative list of products based on their mechanism of action is also given and few potential opportunities for the future are suggested.

**Keywords:** anti-pollution; skin barrier; antioxidants; pigmentation; dark spots; lentigines; aging; fine lines; wrinkles; inflammation

---

## 1. Introduction

"Anti-pollution" is one of the recent buzz words in Personal Care and Cosmetics industry and "Anti-pollution" products are witnessing significant growth in the last couple of years due to combination of several factors some of which are:

- worsening air quality particularly in large metro cities in Asia as well as in the West;
- growing urbanization and increasing visibility of pollution;
- increasing awareness and concern about the adverse impact of environment pollution on health, wellness, and beauty;
- better scientific understanding of impact of environment pollution on skin;
- development of many new active ingredients to protect skin from the adverse impact of environmental aggressors;
- launch and promotion of many products claiming wide range of benefits; and,
- development of hybrid/multifunctional products leveraging the existing habits to deliver anti-pollution benefits.

## 2. Worsening Air Quality and Growing Importance of Anti-Pollution Products

Pollution has long been a consumer concern in Asia due to high level of pollution in cities like Shanghai and Beijing in China, New Delhi and Bangalore in India, and Karachi in Pakistan. Although, pollution level is not high in Europe and North America, pollution is now making headlines in west

and recently pollution level across continent of Africa has also increased significantly [1]. As a result, though the anti-pollution movement first began in Asia Pacific region, it is finally moving into western markets also and hundreds of commercially successful products can be seen in nearly every major market across the world [2].

High pollution levels were initially confined to industrialized and urban areas, however it is now just not restricted to big cities and air quality has taken hit across the board. According to a recent WHO report, only 12% of the people living in cities that are reporting on air quality are exposed to environment where the air quality complies with WHO air quality guideline levels. About half of the urban population being monitored is exposed to air pollution that is at least 2.5 times higher than the levels WHO recommends [3]. Today 54% of global population is urbanized, as compared to just 34% in 1960, and by 2050, it is projected that 6.4 billion people will live in large cities [4]. Recently, a very interesting report with a list of most wrinkle prone cities in 2040 has been published [5].

Various consumer surveys undertaken by different research agencies and various companies show varied numbers regarding consumer awareness and concern about the impact of pollution. However, an overview of all the studies clearly point towards growing importance and a huge potential of anti-pollution cosmetics across the world.

More than 80% of consumers worldwide think, skin absorbs pollution from air. Pollution, dirt, and chemicals from vehicle emissions, plants, factories, cigarette smoke, etc. are seen by consumers as one of the largest causes of skin problems [6]. According to a recent survey, 19% of US consumers, 36% European, and 37% Asian consumers identify pollution as a major source of aggression on skin. Among the Asian consumers, the highest concern for deleterious impact of pollution is witnessed by Chinese consumer, and 41% say they are worried about pollution. German, Spanish, and French women aged 45–54 feel concerns for environmental agents adversely affecting their appearance, the highest concern being expressed by French women [7].

According to another survey, 79% of all the global new skin care products marketed in Q1 2016 brandished an Anti-pollution claim [8]. 38% of new products between January and October 2016 in this category were in Asia-Pacific market [9]. China is the APAC leader of new anti-pollution skincare product launches in 2016, and has remained in the top position in Q1 2017 [10,11].

There has been a sharp increase in searches for the term "Anti-Pollution skin care and formulations". The ingredients that power these anti-pollution products are also seeing a boost in their search volume [12]. It is believed that anti-pollution skin care is beyond a trend: it is a real concern that is here to stay [13].

In China, although 61% consumer report "feeling very concerned" about atmospheric pollution, just 17% of consumers think anti-pollution products are "very useful" and only 25% have bought anti-pollution skin care product. The data for India also show a similar gap between awareness, concern, and actual usage of anti-pollution products [14]. This gap not only indicates a huge untapped market opportunity, but also poses a complex technical challenge to develop efficacious product that deliver the desired benefit effectively and convincingly.

There are several factors that make development of globally harmonized anti-pollution products a massive challenge. The range of pollutants and environmental hazards that affect consumers in different parts of the world is incredibly diverse, with wide variations occurring even within the smallest country [15]. The differences in skin types according to ethnicity, variation in seasonal weather pattern, differences in benefit expectations and variances in personal care, and cosmetics usage habits of consumers in different regions of the world makes the task of developing globally harmonized anti-pollution products very challenging [16].

The first step towards development of efficacious products is better understanding of the various environmental aggressors and mechanism by which they impact skin. A study of different active ingredients available, their mechanism of action can also help in active selection. A review of various anti-pollution products currently available in the market, different benefits claims, and better consumer

insights can also be very useful for developing a comprehensive approach for formulating and marketing of new products.

## 3. Understanding Impact of Pollution

The sources of environment pollution are diverse, including vehicular traffic and exhausts, coal burning power plants, industrial combustion, cigarette smoke, indoor domestic kitchen cooking fires, and Volatile Organic Compounds. The pollutants in these sources include Particulate Matter (PM), oxides of carbon, sulfur and nitrogen, ozone, free radicals, and other airborne chemicals, like pesticides, chemical sprays, and hydrocarbons [17].

Many in vitro and in vivo studies have been carried out by different researchers to examine the effect of these different environmental aggressors on skin and a summary of the findings from some of these studies is given below:

The information on how exactly air pollution could damage skin is limited. However, there are some studies that have begun to investigate the issue, and some consensus has been reached about the various types impact of air pollution on skin. Self-evaluation by consumers has confirmed that skin quality is impacted by the bad environmental conditions and depending on skin type, people observe an aggravation of their skin problem e.g., dry and dull skin, dark spots and uneven skin tone, wrinkles and fine lines, oily skin and acne, sensitive skin, and imperfection [18].

Most of the experts agree that pollution can damage skin barrier, result in depletion of vitamin E and squalene level, and breakdown of collagen and elastin exacerbating existing skin problems such as dehydrated skin, hyperpigmentation, photoaging, excessive sebum secretion, inflammation and sensitive skin, eczema, and atopic dermatitis [19].

Skin's protective proteins—keratin in the outer layer and collagen in the lower layer-guard against moisture loss and maintain skin's elasticity. The defensive capacity that is provided by keratin and collagen is limited and if the level of air pollution becomes too high, it could potentially overload these protective proteins resulting in disturbances in or even damage to skin's protective structure [20].

The formation of dark spots, also called lentigines, increases with levels of traffic-related air pollution and soot particles from traffic pollution are associated with a more pronounced nasolabial fold resulting in increased visible aging effect.

Studies conducted in vivo, under real life conditions of exposure have shown an increase of production sebum and composition of sebum is also modified [21].

## 4. Mechanism of Skin Damage

Skin exposed to pollution areas experiences a higher sebum secretion rate and higher amount of lactic acid, resulting in a decrease in subcutaneous pH when compared to non-polluted areas. Cholesterol, Squalene, and vitamin E, which are the main antioxidants at the surface of the skin are decreased in polluted areas, as these antioxidants are mobilized to combat oxidative stress in the skin. This results in lower ratio of squalene/lipids, poorer cohesion of stratum corneum, and higher erythematous index (redness).

Some of the parameters that can be used as potential anti-pollution markers are skin pH, level of vitamin E, lactic acid, elastin, collagen, and skin lipids, such as triglycerides, free fatty acids, squalene, wax esters, and cholesterol, which are believed to decrease with pollution. The other parameters, such as the sebum secretion rate, and the levels of glycosylation end products, malondialdehyde, squalene monohydroperoxide, oxidized proteins, interleukin IL1a, and adenosine triphosphate increase with pollution.

The environmental pollutants seem to act according to a common mechanism involving the Aryl Hydrocarbon Receptor (AhR) that can be found on several types of skin cells, such as keratinocytes, fibroblasts, melanocytes, and Langerhans cells. It is believed that it is the activation of this receptor under the effect of various environmental aggressors that triggers in cells the expression of various genes controlling the reactions related to oxidative stress, the induction of pigmentation or

inflammation, immunosuppression, and premature aging. Atmospheric antagonists can cause their harmful effects via their impact on the function of key enzymes and biocatalysts within the human body [22].

Air pollutants consists of a heterogenous mixture of PM and organic substances, like Polyaromatic Hydrocarbons (PAH), such as benzo[a]pyrene (BaP) that is bound to the surface of these particles. Even simple surface interactions can drastically alter the composition of the skin, influencing surface barrier function, increasing Trans Epidermal Water Loss, and compromising skin hydration. In addition, these pollutants are capable of penetrating into deeper layers of epidermis and can adversely impact the skin especially because its physical properties make it highly reactive towards surfaces and biological structures.

PM also adversely impacts expression of hydrophobic epidermal lipids including cholesterol sulfate, phospholipids, sphingomyelin, and glucosylceramides, which are key components for barrier function. PM also affects expression of Claudin-1 which is an important protein involved in the integrity of tight junctions.

PM in pollutants can lead to an increase in cytokine levels. Lung mRNA levels of antioxidant/phase II detoxifying enzymes also decreases due to exposure to the PM. The exposure to PM also reduces mRNA levels of involucrin, transglutaminase-1, and E-cadherin, which are the markers of skin barrier function and play a role in skin cohesion and mechanical resistance to stress. The level of enzyme Caspase 14 that is required for filaggrin degradation into natural moisturizing factors (NMF) also decreases due to the action of PM.

The cytoplasmic transcription factor AhR is activated in the presence of pollutants that induce the expression of genes responsible for changes in the barrier function, in melanogenesis and inflammation, in addition to inducing oxidative stresses in human skin. This results skin dehydration, pigment spot formation, and in increased signs of extrinsic aging, including wrinkle formation and loss of firmness and elasticity [23].

The fine dust particles initially deplete skin of antioxidants and leave skin feeling dry. Continued exposure may lead to premature wrinkles and larger than normal pore size, and, in some cases, result in oily skin. Air pollutants, PM, and PAH enter the skin and generate quinones, which are redox-cycling chemicals that produce Reactive Oxygen Species (ROS). The PM increases the amount of ROS that triggers the upregulation of metalloproteinases which in turn leads to extrinsic aging and skin pigmentation. The exposure to Poly Aromatic Compounds enhances the level of oxidative stress and genotoxic damage, and thus contributes to skin aging process [17].

It was also observed that metals present in PM are largely responsible for the observed oxidative stress. Metals promote ROS production and cause antioxidant response element (ARE) promoter activation. ARE is a cis-acting enhancer sequence that is found in the promoter region of many genes encoding antioxidant and Phase II detoxification enzymes/proteins. In response to oxidative stress, regulatory proteins, such as NRF1, NRF3, and BACH1 bind to AREs and compete for binding with NRF2. NRF2 mediates a transcriptional network of responsive genes that modulate in vivo mechanisms against oxidative damage and reactive electrophiles.

Ozone disturbs stratum corneum lipid constituents that are known to be critical determinants of the barrier function. Ozone speeds up skin ageing by depleting vitamin E levels in the skin, interfering with wound-healing processes, and causing oxidative stress. Ozone has also been found to influence lipid peroxidation levels, which, could in turn, lead to additional additive effects of the other stressors like Polycyclic Aromatic Hydrocarbon.

The results of a recent study linked the formation of dark spots on the skin—known as lentigines—with levels of traffic-related air pollution and more particularly to the level of $NO_2$. The study also showed that the most significant changes were revealed in cheeks of Asian women and $NO_2$ had a slightly stronger effect than PM concentration [24].

PM induces oxidative stress, production of ROS, and secretion of pro-inflammatory cytokines, resulting in lipid peroxidation and DNA damage. This further increases matrix metallo proteinases,

(MMPs)-1, -2, and -9, which degrade collagen. In a separate study, over a five-year period, ozone was strongly linked to wrinkle formation due to antioxidant and collagen depletion [25].

Polycyclic aromatic hydrocarbons, commonly found in air pollution, are known to also induce an inflammatory response. AhR activation potentiates inflammation, which eventually overwhelms the skin's natural defenses depleting its antioxidant capacity and contributing to a background inflammatory environment and disrupting the skin barrier function.

It is observed that exacerbation of acne due to pollution is linked to inflammatory processes, and while the prevalence of acne is similar between Asian and Caucasian women, Asian women experience inflammatory acne more frequently and report flare-ups during high pollution periods. This review revealed that acne is believed to have multiple causes based on three main observations: increased sebum production, abnormal keratinization, and, more recently, inflammatory reactions [26].

Ozone has been shown to increase cytochrome p450 activation via the AhR. Once in the skin, the ROS causes a lipid peroxidation cascade, which stimulates the release of other proinflammatory mediators from keratinocytes and melanocytes, thus establishing a vicious inflammatory cycle, with infiltration of other potentiating cells such as neutrophils and phagocytic cells.

## 5. Formulating Anti-Pollution Products

Pollution affects skin a several different ways and it is difficult to develop globally harmonized products due variations in level and type of pollution, difference in skin type according to ethnicity, as well as different benefit expectations by consumers in different parts of the world. However, based on the above understanding of nature of pollutants and mechanism by which they damage skin, generic guidelines for formulating new products can be developed. It is also difficult to develop a single product that can deliver several different benefits and ideally a regime involving the use of multiple products may be necessary.

A general strategy for product development is a routine including topical products that remove, repair, and protect. Most products currently available in the market counter the effect of pollution in one or several of the following manners: (Figure 1)

- reduce particle load on skin by cleansing or exfoliation;
- prevent deposition and penetration of pollutants on skin;
- restore and strengthen the skin's protective barrier structure and function;
- reduce trans epidermal water loss and thus improve skin hydration;
- replenish antioxidant reserves;
- reduce inflammation;
- control melanogenesis;
- promote collagen/elastin synthesis; and,
- protect skin from harmful UV rays which exacerbate effect of the other environmental pollutants.

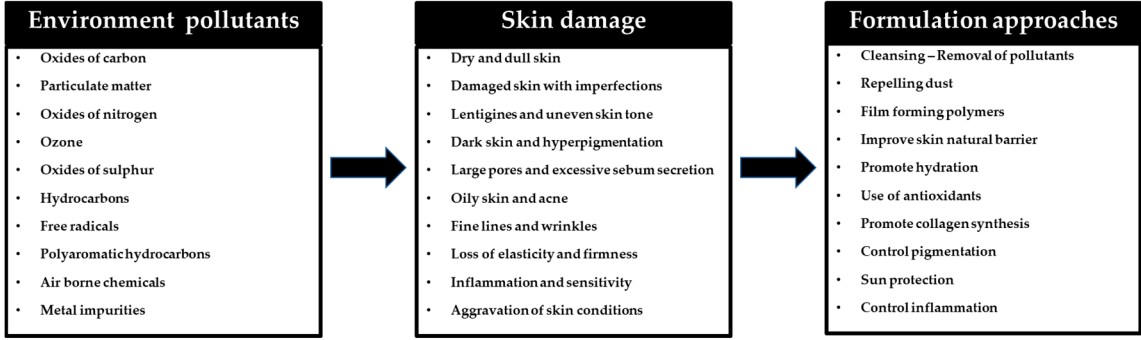

**Figure 1.** Environment pollutants, skin damage and guidelines for produce design.

In addition, any solution aimed at alleviating the effects of pollution must address skin sensation as well as appearance.

Chinese products are relatively more focused on hydration and whitening claims, whereas Japanese brands stand out on UV protection, brightening and hydration claims. South Korea is more active with natural and "free from" products and overall Southeast Asian markets have relatively higher portion of male targeted products. In Europe and North America, products that are fortified with antioxidants and vitamins claiming skin strengthening, repair, anti-inflammatory, and anti-aging benefits draws more traction, and features like "organic" and "environmentally friendly" are preferred particularly in Europe. LATAM and Southeast Asia are more active in the cleanse and protect space [16].

Also, one of the growing trends in cosmetic industry is the use of multifunctional hybrid products and this offers ideal opportunity for entry of anti-pollution products/benefits by tagging with existing products offering anti-aging, skin brightening, and sun protection benefits. Anti-pollution benefits can also be offered in different product formats like make up powders, facial wipes, masks, mists, and exfoliating products.

## 6. Active Ingredients and Products

In the following section, a summary a few select active indigents that can be used in Anti-Pollution products, classified based on their mechanism of action is given. The list is not exhaustive and there are many other active ingredient options that are available in each of these categories. Many ingredient suppliers have developed materials that are a mixture of actives with different mechanism of actions, and as a result, some of the actives given below may fit in several different categories of actives.

### 6.1. Cleansing and Use of Film Forming Polymers

The first approach to alleviate or prevent impact of environmental aggressors is to control deposition and penetration of PM on skin and to remove the deposited or penetrated pollutants. There are several surfactants and barrier-forming polymeric materials that can be used in different product formats, like cleansers, masks, exfoliators, moisturizers, and even wipes and mists. The surfactant selected should be such that it is not harsh on skin and the use of such products should not result in increasing the penetration of pollutants in the skin pores.

Activated Charcoal with high porosity helps to draw toxins from pollution up and out of the skin [27]. Solid adsorbent materials, like Kaolin, Magnesium Aluminum silicate, and scrubs, like coffee beans and Rice Bran, are also used in many anti-pollution products, like masks and exfoliators. PEG 20 Glyceryl Triisostearate is reported have the right Hydrophilic-Lipophilic Balance to effectively remove PM from skin, and it is also a mild surfactant [28].

Biosaccharide gum-4, an anionic, deacetylated branched polysaccharide of high molecular weight, forms non-occlusive barrier and has matrix-forming properties that protects skin against different environmental aggressors [29].

Brassica Campestris and Aleurites Fordii Oil Copolymer is made from Tung (China wood) and Rapeseed oil. It forms a flexible and uniform film on the skin for even coverage and has the power to stay on skin longer, delivering superior protection from damage that is caused by environmental aggressors. In a clinical anti-pollution study, treatment with this active showed a 32% improvement in the protection from microparticles [30].

An active ingredient blend consisting of Lecithin, Acrylic Acid/Acryl amidomethyl Propane Sulfonic Acid Copolymer, Dimethyl methoxy Chromanol, Xanthan Gum, Glyceryl Caprylate, and Diisopropyl Adipate provides a barrier between the skin and the harmful ambient substances and can help prevent the accumulation of pollution particles in epidermis. In addition, metal chelating properties of various ingredients in the blend also increases its capacity to resist harm caused by heavy metals [31].

A few examples of products that use this approach for delivering anti-pollution benefits are Pond's Men Pollution Out Face Wash with a combination of Charcoal and Coffee Bean Scrub, Bioelements

Advanced Vita Mineral Deep Detox Mask with Activated Charcoal, Cosmedix Deep Cleansing mask with a combination of Kaolin clay, salicylic acid and Sulphur, Tata Harper's Purifying Cleanser and Mask with Kaolin and Magnesium Aluminum Silicate, and Etude House Dust Cut Facial Mist with shield forming polymer.

*6.2. Strengthening Skin Barrier and Improving Hydration*

Pollution damages or weakens the skin's natural barrier resulting in higher and deeper penetration of pollutants, which, in turn, affects the morphology and integrity of skin structure. The damaged barrier also leads to higher trans epidermal water loss, resulting in poor skin hydration affecting the rigidity and firmness of skin. One of the approaches to counter the impact of environmental pollutants is to strengthen the skin's natural barrier and improve the hydration. A few examples of active ingredients in this category are listed below:

A mixture of skin identical long chain and short chain ceramides, particularly with skin identical stereochemistry along with vegetable based cholesterol and behenic can create a multilamellar system resembling the structure of lipid barrier in the stratum corneum. It can revitalize damaged skin by optimizing the total epidermal water management system i.e., repairs skin's own protection barrier, activates skin's natural water moisturizing system (sphingolipids, filaggrin), and is ideally suited for use in anti-pollution products [32].

Selaginella Lepidophylla (Rose of Jericho or the Desert Rose) contains complex enzymes and stress response elements that work synergistically to prevent significant damage during long period of desiccation and work to promote repair during rehydration and hence the plant is also known as "Resurrection Plant". The extract of this plant containing highly adaptive Moisture Retention Complex with film forming properties improves skin barrier function, offering intense moisturizing benefits and reducing trans epidermal water loss from compromised skin. In addition, it also inhibits accumulation of PM on skin, provides antioxidant benefits, and enhances cellular proliferation [33].

The extract of Tremella Fuciformis containing polysaccharide has an intense moisturizing and light film forming properties. It protects skin from pollution by forming a soft hydrating film and provides superior moisturizing effect, and is found to be more effective than hyaluronic acid [34].

The extract of Leontopodium alpinum "Helvetia" Edelweiss, contains active compound Leontopodic acid which strengthens the epidermal barrier and reduces skin sensitivity. It stimulates several key genes and proteins responsible for epidermal protection, including transglutaminase 1, involucrin, loricrin, and keratins. It stimulates key proteins that are responsible for providing protection and moisturization of the skin, and significantly improves skin barrier integrity and stratum corneum cohesion, leading to skin that is better protected and more resistant environmental aggressors [35].

The extract of Chondrus Crispus (Red Algae), develops shield like film to protect skin from environmental stresses due to the high level of carrageenan that is found in this alga, and is also known as a "Second Skin". It acts as an instant film former due to high level of polysaccharide providing on the spot hydration along with a very smooth texture. It protects skin against dryness and irritation caused by pollution by replenishing and maintaining skin's natural moisturizing factor and helps reduce Trans Epidermal Water Loss [36].

A few examples of products that use this approach for delivering anti-pollution benefits are Paula's Choice Resist Omega Plus with Omega Fatty Acids, Bioelements Remineralist Daily Moisture with Red Algae extract, and Sea Salt Minerals, Jurlique Rose Moisture Plus Moisturizing Cream Mask with Rose Hip Oil, Clarins Hydra Essential range of products with Kalanchoe Pinnata extract, and Decelor Hydra Floral Anti-Pollution Hydrating range with Moringa Oleifera extract.

*6.3. Use of Anti-Oxidants*

PM and other environment pollutants can induce oxidative stress and produce ROS, which can damage cell proteins, DNA, and cell membranes. This can lead to tissue damage, which results in wrinkles, fine lines, dehydration, and loss of youthful volume. The use of actives containing

antioxidants and metal chelating agents in formulation can counter this effect and few actives which deliver anti-pollution benefits by this mechanism is given below:

Salvia Hispanica (chia) seed extract contains high levels of phyto-nutrients, including phytosterols, flavonoids, Alpha Lipolytic Acid, Coumaric Acid, Caffeic Acid, and tocopherols, which give it a strong antioxidant capacity, which, in turn, protects the skin from ROS to speed up the skin's repair systems and prevent further damage. Chia oil also significantly increases skin hydration, reduces trans-epidermal loss of water, and improves skin barrier function to minimize fine lines and wrinkles [37].

A plant extract that is derived from Camellia Sinensis, White Tea, is reported to be rich in antioxidant flavonoids, which neutralizes up to 80% of free radicals, protecting cell membranes and helping the skin to defend itself from pollution and other environmental aggressions [38].

Myrtle leaf extract is rich in hydro soluble flavonoids and tannins, resulting in a very potent anti-oxidant and skin protecting properties. Myrtle extract promotes a well oxygenated, stress free complexion capable of neutralizing the effects of pollutants, and free radical damage [39].

The Pink Pepper Solution derived from tree Schinus Molle, (The Tree of Life) is rich in highly active flavanols and phyto antioxidants, quercitrin, and miquelianin. It provides cellular protection against oxidative stresses that is induced by the fine suspended dust particles by controlling antioxidant enzyme activity. It is also shown to be associated with a modulation of epigenetic biomarkers that are linked to a stronger skin barrier and reduces skin permeability that is induced by environmental stress and increased skin hydration [40].

Nature is a constant inspiration for cosmetics ingredients, but plants are not the only sources of natural actives. An innovative active mineral extract rich in trace elements and copper is extracted from malachite using special technology to obtain a unique bioavailable stone active. A copper-rich pollution magnet and heavy metal scavenger, this extract stimulates the natural antioxidant pool, helps support skin's natural cellular damage protection system, and boosts cellular defenses, thus offering protection against environmental oxidative stresses [41].

A few examples of products which use this approach for delivering anti-pollution benefits are CelleCle Celle Surge Intensif serum with Thermus Thermophillus, Wei Beauty Golden Root Multiaction Anti-pollution mist with antiozonate complex, Eminence cleansing oil, Oxygenating Fizofoliant and Detoxifying Overnight treatment with Swiss Cress Sprout extract, stone oil and lotus extract, Yon-Ka's Vital Defence Daily Skin Cream with Organic Myrtle, Coenzyme Q10, vitamin E and Moringa Peptide, and Guinot's Biooxygene Face Serum containing Pro-oxygen.

*6.4. Prevent Degradation of Collagen/Elastin and Improve Skin Firming*

Pollution is also known to degrade the structural proteins like collagen and elastin, and as a result, the skin firmness and elasticity is adversely affected resulting in extrinsic signs of aging. Actives, which can counter the degradation of collagen and elastin, can improve the efficacy of anti-pollution products and a few examples of actives that deliver anti-pollution benefit by this mechanism is given below:

An extract of microalgae Nannochloropsis Occulata, rich in vitamin C, vitamin B12, and polysaccharide-Pullulan, supports the repair and maintenance of the extracellular membrane for a superior skin firming effect. It forms a thin film on the skin, exerting an instant perceptible tightening effect. It also helps to stimulate the formation of collagen I, which is an essential part of the skin's connective tissue, and as a result, actively tightens and firms skin in the long term [42].

Paeonia Albiflora root extract, rich in oligosaccharide fights the premature aging by regulating cellular communication. It provides gradual replumping, volumizing and firming by controlling communication and limiting negative pro-inflammatory exchanges between the dermis and hypodermis, and increasing the thickness and volume of the adipose tissue [43].

An extract of Undaria Pinnatifida—a kelp Japanese Sea Algae, contains an active ingredient that is called sulfated polyfucose, which protects cell wall integrity and stability against damaging

environmental factors. It maintains skin firmness, elasticity, and smoothness, supports skin regeneration, and defends against environmental stress. The extract also contains antioxidants, vitamins, protein, and is also rich in vital minerals, like calcium, iron, potassium, and sodium. It is also a rich source of eicosapentaenoic acid, an omega-3 fatty acid, and of polysaccharides. It offers several other anti-pollution benefits to protect skin against environment pollutants by different mechanisms [44].

White Tea extract, according to Professor Declan Naughton at Kingston University, protects the structural protein of the skin. The study showed that white tea extract reduced the activities of the enzymes, which breakdown elastin and collagen and improve skin's natural elasticity and strength [45].

An extract of Astragalus Membranaceus Root, Atractylodes Macrocephala Root, and Bupleurum Falcatum Root is shown to target particularly the newly discovered Yin-Yang-1 protein to adjust the epidermis maturation and restore the epidermis integrity, stimulate collagen synthesis, and counteract the melanogenesis and the oxidative stress from UV radiations and pollution [46].

A few examples of products that use this approach for delivering anti-pollution benefits are CelleCle 3D Strata Sculpt Remodeling Serum with Peony Root Oligosaccharide, Ren V-Cense Youth Vitality Day Cream with SAP proteins from Orange Stem Cells, Frankincense and Boswellic Acid, Erno Leszlo Firmarine Hydrogel Mask with Argania Spinosa Kernel oil, Wei Dan Gui Deeply Nourishing Sheet Mask with Astragalus extract and Arcona Desert Mist with Mucin, Glutathione, and Marjoram extract.

*6.5. Control Pigmentation, Skin Lightening Reduce Dark Spots*

Pollution affects the melanin synthesis pathways and can result in dark spots, overall skin darkening, and uneven skin tone and increase extrinsic signs of aging. Actives that regulate melanin synthesis pathways can counter the impact of various environment pollutants and a few examples of active that act by this mechanism of action are given below:

An extract of Lepidium Sativum (Swiss Garden Cress) Sprout contains sulforaphane, a powerful antioxidant phytonutrient. It effectively inhibits pigmentation by targeting two key upstream reaction steps in the melanin cascade. It neutralizes ROS and inhibits α-MSH, a natural hormone, which stimulates skin pigmentation. It provides exceptional brightening benefits, fades the appearance of dark spots and discolorations, evens skin tone and is particularly found to be effective on Asian skin type [47].

Nature identical resveratrol decreases melanin synthesis through different pathways, e.g., it inhibits Tyrosinase, regulates MCR1 (α-MSH receptor), affects the function and maturation of melanosomes, attenuates melanosome transport within melanocytes, as well as the melanosome transport to keratinocytes. It has excellent skin lightening properties, gives radiant and youthful looking skin, and visibly brightens skin [48].

An aqueous complex containing extracts from Swiss Alpine plants—Malva Sylvestris (Mallow), Alchemilla Vulgaris, Melissa Officinalis, Mentha Piperita (Peppermint), Veronica Officinalis, Achillea Millefolium and Primula Veris contains Flavonoids, Phenolic Acids, Polysaccharides, and Iridoids. It shows tyrosinase inhibiting activity and reduces the color intensity of dark spots, provides skin lightening, and evens skin tone [49].

An extract of 10 Chinese whitening herbs—Panax Notoginseng root, Gastrodia Elata root, Poria Cocos, Glycyrrhiza Uralensis root, Panax Ginseng root, Carthahus Tinctorius Flower, Salvia Miltiorrhiza root, Paeonia Suffruticosa Root, Scutellaria Baicalensis root, and Lycium Chinese Fruit is found to inhibit melanin synthesis, maturation, transport, operation, and degradation to achieve healthy whitening [50].

A marine exopolysaccharide isomerate containing two amino acids, serine, and alanine can fix the melanocyte receptor and prevent formation of the pigmentary synapse. It has been clinically proven

in both, in vitro and in vivo tests to reduce pollution induced melanin synthesis, significantly reduce dark spots, and improve skin tone with uniform pigmentation [51].

A few examples of products which deliver anti-pollution benefits by this mechanism of action are SkinCeuticals Phloretin CF with Phloretin and Ferulic Acid, Anne Semonin Brightening Serum with Sea Flower and Brown Algae extract and Dr Dennis Gross Dark Spot Sun Defense Broad Spectrum SPF 50 Sunscreen with Melatonin Defense complex.

## 6.6. Reduce Inflammation

PM induces secretion of pro-inflammatory cytokines, such as TNF-$\alpha$, IL-1$\alpha$, and IL-8, which leads to the increased skin inflammation, redness, and skin aging. Therefore, the incorporation of anti-inflammatory actives in formulations can be very useful in alleviating some of the adverse impact of environment pollutants. A few examples of actives that act by this mechanism is given below.

An extract of Paeonia Lactiflora (White Peony) roots is known in Traditional Chinese Medicine for its healing properties and it also calms nervous irritability. This extract provides protection against pollution and urban stressful environment. It also acts on the uniformity of skin color by correcting excessive redness and reducing ageing spots [52].

E/Z2benzylidene5,6dimethoxy3,3dimethylindan1one (BDDI), temporarily binds to the AhR to block it for PAH. BDDI significantly reduces the Proopiomelanocortin (POMC) gene expression that is induced by Diesel Exhaust Particle (DEP). POMC released from keratinocytes triggers melanin biosynthesis and is responsible for the formation of dark spots. BDDI also significantly reduces the DEP-induced expression of inflammation gene markers, such as Interleukin-6 (IL-6), which is linked to an inflammatory response of the skin (inflammaging) [53].

Zingiber Officinale (Ginger) Root Extract is used as an anti-irritant, and it improves skin texture. It enhances the texture and smoothness of stressed skin, smoothing wrinkles, and complexion regularity. Ginger´s activity is mainly based on its action on the arachidonic acid pathway, resulting in several actions, such as anti-inflammatory, analgesic, antipyretic, and it is known to fight redness and irritation and provide soothing effect [54].

An extract of Rubus Idaeus (American red raspberry) is found to exhibit anti-inflammatory activity, mainly due to the presence of high levels of flavonoids and anthocyanins. In addition, it provides antioxidant power due to the presence of natural phenolic components, and induces genes responsible for DNA protection and repair [55].

An extract of Asteriscus graveolens, Arabian Desert Daisy has been found to protect skin cells from oxidative stress, shielding it against damage from pollution. It protects cells from air pollution induced cell death by reducing the expression of cell death related genes and stimulating expression of phase I enzymes, such as cytochrome p450, thereby increasing cell survival. It reduces signs of premature aging and protects cells from inflammatory response [56].

A few examples of products which use this approach for delivering anti-pollution benefits are Koh Gen Do Oriental Plant lotion I with Tremella Frucformis, Erno Leszlo Sensitive Hydrogel Mask with Honey Suckle extract, Dermalogica Ultracalming Cleanser with Lavender, Echinacea, Raspberry and Cucumber extract and Skin Medica Sensitive cleanser with Chamomila Recutita Flower, Cucumis Sativus, and Calendula Officinalis Flower Extract.

## 6.7. Use of Multifunctional Actives or Combination of Different Actives

Pollution damages skin in many ways and to alleviate impact of pollution a combination different active ingredient may be necessary. Several active ingredient developers market a mixture of actives which counter the impact of pollution by different mechanisms and there are many actives which provide multiple benefits by acting in different ways. A few examples of such multifunctional actives or blend of actives is given below:

Moringa Oleifera Seed Extract from "tree of miracles "or "never-die tree" shields the skin from pollution in many ways. The peptide rich extract possesses highly protective and purifying

properties. Its antimicrobial action eliminates the asphyxiating microparticles that are produced by the environment, so the skin is purified and can benefit from optimum oxygenation. It protects DNA from the damaging effects of pollutants and heavy metals. The extract also increases collagen synthesis, improves cellular viability, and offers skin conditioning and nourishing benefits, resulting in enhanced skin complexion and healthy glow [57,58].

Passiflora Edulis Seed Oil (Maracuja) is shown to repair skin tissue at three levels by stimulating the key markers of cell proliferation, cell migration and the Dermis Epidermis Junction. It also regenerates, restructures, and remodels the skin by stimulating the synthesis of elastin and collagen and boosts the contractile force of fibroblasts. It repairs, regenerates, and soothes skin that is weakened and damaged by the actions of different environmental stressors [59].

Passiflora Edulis Fruit Extract which is rich in polyphenols harnesses the hormesis pathway by activating the natural self-defense systems and preventing the deleterious effects of In and Outdoor pollutants by free radical scavenging. It detoxifies and protects skin from the strong stresses of environment pollutants and restores and prevents the degradation of the essential components of the skin barrier and extracellular matrix preserving the barrier function, structure, and mechanical properties of the skin and the aging process is slowed down [60].

The extract of Marrubium vulgaree (Horehound) is shown to protect skin cells from the penetration of pollutants while fighting free radicals. It has also been shown to support the removal of toxic oxidant species and fight oxidative stress, while protecting, strengthening, and repairing skin and reducing inflammation [61].

Ascophyllum Nodosum Extract is a concentrated form of Fucoidans (sulfated polysaccharides) of high molecular weight mainly made up of fucose and it also contains xylose and glucuronic acid. Cytoplasmic transcription factor, AhR is activated in the presence of pollutants and it regulates numerous cell functions, such as inflammation, cause changes to the barrier function and melanogenesis. The extract inhibits the AhR pathway and impacts Dermo-epidermal restructuring, controls the appearance of dark spots and reduces formation of redness [62].

An extract of Camellia Japonica Flower (Red Snow or Rose of Winter) has anti-oxidant effect, anti-irritation and collagen boosting efficacy, increases skin hydration, and shows anti-wrinkle efficacy [63].

A mixture of Salvia Hispanica Seed Extract, Trehalose, Galactoarabinan, Glycerin, Xylitol, Sodium Phosphate, and Sorbitol is reported to provide defense against harmful polluting agents. It reduces permeation of polluting agents by 65%, protects the DNA of cells inhibiting premature aging, inhibits hyperpigmentation, and provides anti-inflammatory properties by reducing Interleukin-6 by 37% [23].

A few examples of products that use this approach for delivering anti-pollution benefits are Bioelements Remineralist Daily Moisture with Sea Salt Minerals, Red Algae, and Malachite extract, Laniege All Day anti-pollution Defensor Serum with extract of Saururus Chinensis, green tea and ginger oil, Clarins UV plus Anti-Pollution SPF 50 Broad Spectrum Sunscreen with White Tea extract, Cantaloupe Melon and Blackcurrant extract and Sisley Phyto-Blanc Brightening Daily Defense Fluid with Stabilized vitamin C, Hexyl Resorcinol, Modified Rhubarb extract, Buckwheat seed extract, Ascorbic Acid 2-glucoside, Provitamin B5, and Phyto squalene.

## 7. Summary

The task of designing effective anti-pollution products is very challenging due to complex nature of the various environment pollutants, different damages that these pollutants cause on skin, and a wide range of actives with different mechanism of action that is available to choose from. The various aspects of pollution that is discussed in the previous sections viz. nature and impact of various environmental pollutants on skin, their mechanism of action, various actives available to alleviate the effect of pollutants, and a few examples of product offering different benefits can be used for developing an overall approach and selection of actives for designing new products. The broad

formulation development approach and guidelines for selection of actives depending on the type of skin damage is summarized in Table 1.

**Table 1.** Guidelines for formulating anti-pollution products.

| Sr No. | Visible Skin Damage | Formulation Approach | Active Options |
|---|---|---|---|
| 1 | Dull and oily skin | Deep Cleansing<br>Exfoliation<br>External polymer barrier<br>Dust repellent polymer | Mild surfactant<br>Activated charcoal<br>Coffee beans and rice bran scrub<br>Biosaccharide gum |
| 2 | Dry and damaged skin | Restore natural lipid bilayer<br>Strengthen skin's natural barrier | Long and short chain ceramides<br>Cholesterol and behenic acid<br>Extract of Edelweiss<br>Extract of Red Algae |
| 3 | Dehydrated rough skin | Improve skin hydration<br>Reduce TEWL<br>Replenish NMF in skin | Extract of Desert Rose<br>Extract of Tremella Fuciformis |
| 4 | Wrinkles and fine lines<br>Loss of youthful volume | Control formation of ROS<br>Use metal chelating agents<br>Replenish antioxidant reserve | Chia seed oil<br>Pink Pepper extract<br>Extract of Malachite<br>White Tea extract |
| 5 | Uneven skin tone<br>Skin darkening<br>Formation of lentigines | Control Melanin synthesis<br>Inhibit Tyrosinase<br>Regulate melanosome transfer | Nature identical Reservatrol<br>Extract of Swiss Garden Cress<br>Marine exopolysaccharide isomerate<br>Extract of Chinese whitening herbs |
| 6 | Loss of skin firmness<br>Loss of elasticity | Promote collagen/<br>elastin synthesis<br>Prevent degradation of proteins | Extract f Nannochloropsis Occulata<br>Paeonia Albiflora root extract<br>White Tea extract<br>Extract of Japanese Sea algae |
| 7 | Skin redness and<br>sensitivity<br>Inflammation and acne | Autoinflammatory actives<br>Use of skin soothing agent | Extract of White Peony<br>Ginger root extract<br>Extract of American Red Raspberry<br>Extract of Arabian Desert Daisy |

In addition to delivering targeted benefits it is very important to deliver the right sensory properties to create the first impression immediately on product use. By understanding drivers behind emulsion aesthetics, it is possible to engineer a complete sensory experience that supports overall product positioning and claims. The other important aspects to be kept in mind while formulating product is claims, consumer, and clinical testing in support of the claims and applicable regulatory guidelines.

The issues that are associated with pollution damage are numerous and varied, ranging from dryness, dark spots, fine lines and wrinkles, dullness and uneven skin tone, loss of firmness, inflammation, and aggravation of acne, and there is no agreement on which pollution bio-markers are best to assess the efficacy of anti-pollution products. There is no international guideline on standardized anti-pollution tests and different types of studies are currently available and used by companies to demonstrate claims [64]. Many brands rely on supplier data [65], and some of the brands even use consumer test data to support their claims. However, as awareness and understanding about pollution is increasing, in vitro and in vivo clinical test data would be useful to differentiate product in a highly competitive and crowded market space with many new products launches every year, and to establish competitive edge in a credible and convincing manner.

### 8. Future Opportunities

The understanding about various environmental pollutant and their mechanism of action is continuously evolving, and ingredient developers are introducing many new actives every year. The growing palate of actives is offering wider choices to the formulators to allow for them to create more efficacious products and the anti-pollution market is likely to grow significantly for many years to come. The new methods of in vitro and in vivo testing will also result in brands making stronger claims, increasing consumer awareness, and generating more demand pull.

One of the growing trends in skin care is the use of multifunctional products as consumer are seeking to simplify their skin care regime. Anti-pollution products can tap into well-established global consumer demand for anti-aging, skin lightening, and sun care products. Some of the product formats like facial mist, wipes, exfoliating products, and compacts and loose powders can be very useful formats for delivering anti-pollution benefits. Anti-pollution benefits are now expanding beyond skin care ranging into hair care, eye care, and color cosmetics, as well [66].

Interestingly, consumer research shows that awareness about pollution, concern regarding the adverse impact of pollution skin and the intentions to try anti-pollution products is very similar among men and women and this points towards a potential opportunity for brands to develop products that are specially designed for men [10].

Many ingredients manufacturers are being quick to develop antipollution solutions that are also natural or organic, taking steps to pre-emptively cater to the naturals trend rather than retrospectively create alternatives to synthetic active ingredients, as has so far been necessary for other areas of the industry. Some of the other trends that anti-pollution products can leverage are rising consumer interest in health and wellness products, products that combat life style stressors, aromatherapy products for emotional wellbeing, products which are sustainably sourced and do not adversely impact the environment, and products making "free from" claims [67].

The trend "Beauty from inside-out" is receiving a lot of traction from consumers and Beauty Supplements, Nutraceuticals, Beauty Drinks, and Functional Foods should also be evaluated by brands to complement their anti-pollution product portfolio [68].

Devices such as cleansing brushes, which use gentle sonic action to dislodge pollutant and help deep cleanse and devices that can increase product penetration using Intense Pulsed Light technique for more effective delivery of actives at the target site are the other opportunities in the anti-pollution market.

Most brands have skin assessment and product selection tool on their site, which recommends products ideally suited for consumer based on response to a few simple questions on condition of skin and the environment consumers are exposed to. When new anti-pollution products are introduced in the existing product portfolio, modification of the product selection tool to include specific questions related to environment pollutant can lead consumers to the newly introduced products.

Some of the products that are likely to hit the market are wearable devices and patches, indicating the exposure to type and level of pollution with smartphone recommending protection required against pollution. Smart mirrors, which help consumers to track condition of skin and assess the change over time on products usage, are the other opportunities in future.

**Conflicts of Interest:** The authors declare no conflicts of interest.

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
