# Peer review of "Guidelines for Formulating Anti-Pollution Products"

_cosmetics, doi:10.3390/cosmetics4040057_

Round 1

Reviewer 1 Report

This review is entitled "Guidelines for formulating Anti-Pollution Products" and this title suggests that specific "do's and don't's about how to formulate these products will be the focus of the review. But that's not at all what the review is about. It's mostly a summary of ingredients from suppliers that might be useful in an "anti-pollution" product. In fact, from page 5 to the end, the review is simply a list of ingredients from various suppliers and their possible uses. 

The authors mention, in the introduction, that developing global anti-pollution products is a "massive challenge" for a variety of reasons, including variability in the type of pollutants, ethnic variations, etc. It seems that coming up with "Guidelines" to meet this challenge should be the focus of the review.

Alternatively,  this review would be made more interesting if it was a more science oriented review of the mechanisms by which pollutants damage skin. As the authors state, "information on how exactly air pollution could damage skin is limited". However, doing a thorough scientific literature search to look at mechanisms by which various pollutants trigger cellular mechanisms that lead to skin damage (ECM destruction, apotosis, cytokine production, etc.), would be much more interesting to a scientific audience. In this regard, the authors do spend one page, "Mechanism of skin damage", providing a brief overview of the mechanism of action of some pollutants. A more in depth review on this subject would be interesting. Another interesting possibility would be to discuss specific topical formulation designs, containing specific ingredients, that have been proven to either block pollutants from triggering skin damage, or which have been shown to block inflammatory pathways (e.g. block IL-1 or TNF-alpha production) triggered by pollutants. 

As it is, this review simply discusses already well-known effects of environmental pollutants on skin structure and function, including loss of collagen, increase inflammation, hyperpigmentation, etc. 

Author Response

The comments made have really been very useful and changes suggested have been incorporated in the revised draft.

Various mechanisms by which different ingredients of pollutants trigger cellular mechanisms that lead to skin damage (e.g. barrier function, oxidative stresses, pigmentation, aging and wrinkle formation, and inflammation) has been covered greater details in the section 4 – “Mechanism of skin damage “of the revised draft.

Mechanism by which various actives block pollutants from triggering skin damage or are known to block inflammatory pathways is now added to various category of actives covered in the section 6.1 to 6.7.

Information in the revised section 4 and 6 can serve as useful guidelines for formulators to design topical formulation for anti-pollution skin care. 

Reviewer 2 Report

This is an interesting manuscript on the relevant topic of skin protection against pollution-induced damage, a subject that receives global attention. 

1. abstract:

‘Sulphur’: Do not capitalize

'The article summarizes approach for formulation’: change to:  'The article summarizes approaches for formulation'  

2. Language must be improved/corrected throughout the manuscript:

e.g.

introduction:

“Anti-pollution “is one of the recent buzz word in Personal Care and Cosmetics industry”

change to: “Anti-pollution “is one of the recent buzz words in the Personal Care and Cosmetics industry

Sentences such as 'Although, air pollution is not as bad in Europe and North America, pollution is now making headlines in west too and according to a recent report, is proving to be a massive problem across continent of Africa as well. [1]’

are unacceptable, grammar-wise.

3. Style:

Reference to specific researchers should be avoided in the text (.i.e. no names except in quotation): example:

'According to the research expert, Dr Jean Krutmann, scientific studies conducted in vivo 128 under real life conditions of exposure have shown increase of production sebum and composition of sebum is also modified. [21]’

Similarly, journal identity should only be mentioned in the reference itself:

'The results of a recent study reported in the Journal of Investigative Dermatology linked the formation of dark spots on the skin—known as lentigines—with levels of traffic-related air pollution and more particularly to the level of NO2.'

4. Content:

In the context of antioxidant interventions, NRF2-directed strategies should be mentioned (- NRF2, the transcription factor that regulates the cellular antioxidant response, modulated for skin benefit by numerous phytochemicals etc.)

5. Summary: An image/graphic, summarizing the various approaches for pollution-directed interventions would be desirable.

Author Response

All the suggested correction in abstract and introduction have been made in the revised draft.

The paragraph on global pollution level in the section 2 has been modified as suggested.

Reference to researcher or the research institution in the text has been deleted.

The role of NRF2 in fighting oxidative stress triggered by injury/inflammation is included in the section 4 “Mechanism of skin damage “of the revised draft.

A figure giving an overview of various pollutants in environment, extrinsic signs of damage and formulation approaches to counter the skin damage has been added in section 7 - Summary.

Reviewer 3 Report

The theme addressed in the review is interesting, since an increasing number of cosmetic products has anti-pollution claims. However, the authors have not based their review in sound bibliography, and many sections are entirely based in texts found on the web, instead of literature that has been peer-reviewed. Other examples of unsound citations are those based in information provided by materials suppliers.

An unusual way of citing references is often found in the text, such as "According to Dr Fussell, information on how exactly air pollution could damage...".

Author Response

References 1-28 and 67-72 are articles published in various journals like Cosmetics and Toiletries and Global Cosmetics Industry.

Section 6 covers information about various active ingredients and hence most of the citations in this section are based on the information provided by suppliers of active ingredients. However, many of these references cover scientific information about mechanism of action of the actives and results of in-vitro and in-vivo clinical studies.

Reference to researcher or the research institution in the text has been deleted.

Reviewer 4 Report

This review paper is not scientifically sound enough. For example, there is no signal Figure or Table in the paper. Although there are 69 references, a large chunk of them are web pages, which are not peer-reviewed. Therefore recommend rejecting for publication.

Author Response

References 1-28 and 67-72 are articles published in various journals like Cosmetics and Toiletries and Global Cosmetics Industry.

Section 6 covers information about various active ingredients and hence most of the citations in this section are based on the information provided by suppliers of active ingredients. However, many of these references cover scientific information about mechanism of action of the actives and results of in-vitro and in-vivo clinical studies.

Various mechanisms by which different ingredients of pollutants trigger cellular mechanisms that lead to skin damage (e.g. barrier function, oxidative stresses, pigmentation, aging and wrinkle formation, and inflammation) has been covered greater details in the revised section 4 – “Mechanism of skin damage” of the revised draft.   

Mechanism by which various actives block pollutants from triggering skin damage or are known to block inflammatory pathways is now added to various category of actives covered in section 6.1 to 6.7.

Section 6 covers information about various active ingredients and hence most of the citations in this section are based on the information provided by suppliers of active ingredients. However, many of these references cover scientific information about mechanism of action of the actives and results of in-vitro and in-vivo clinical studies.

Round 2

Reviewer 1 Report

The manuscript has been significantly improved by providing more detail about the cellular mechanisms that are affected by pollutants and how these cellular processes damage skin. The authors have provided an acceptable amount of information about this subject to be of interest to readers.

In regard to actual formulation "guidelines" the authors have not improved that area significantly. The review is essentially a listing of ingredients from suppliers that might be useful in anti-pollution products. No discussion is given about HOW formulate an effective product. For example, what percentage of a given ingredient is enough to provide protection? Does the anti-pollutant "active" need to penetrate through the stratum corneum to protect the skin from pollutants, and if so, how is the formulation going to allow the "active" to penetrate? Is the "active" larger than 500 Daltons? If so, it will be useless in a topical product because it cannot penetrate through an intact stratum corneum. Are there certain combinations of "actives" that would be the best to use for an effective anti-pollution product? These are "guidelines" a reader would fine useful and none of this is discussed in this review.

Author Response

A figure summarizing the guidelines for developing overall formulation approach depending on the nature of pollution and skin damage / desired benefit is added in section 5 (Figure 1).  

A new table summarizing guideline for selection of actives / active combination for different skin damages is included in the revised draft in section 7 ( Table 1) .

Some of the actives like dust repellent polymers form a surface barrier and do not penetrate through the stratum corneum. Actives such as ceramides, cholesterol and behenic acid become part of the skin’s lipid bilayer whereas some of the water-soluble actives penetrate and replenish NMF. Some of the actives are used as oils whereas as others are extracted using solvents like glycerine / propylene glycol to facilitate dissolution and penetration of actives.

Particulate Matter (along with materials like BaP adsorbed on the surface) is reported to reside skin pores and some of the antioxidant and metal chelating agents are believed to counter the effect of PM by acting in the skin pores. 

Reviewer 2 Report

The quality of the manuscript has improved significantly.

Some editing for language correction would further improve the readability and stringency of the final product.

example: Sentences such as the following are still part of the revised manuscript and should undergo one more round of editing for language/grammar:

current: 'One of the growing trend in skin care is the use of multifunctional products as consumer are seeking to simplify their skin care regime.'

corrected: 'One of the growing trends (!) in skin care is the use of multifunctional products as consumers (!) are seeking to simplify their skin care regime.'

Author Response

Spelling and grammar check has been done on the entire draft and all language corrections have been made. 

Reviewer 3 Report

The article has been improved, however, in section 4, entire paragraphs have been added without citation of sources.

Author Response

I had covered the different mechanism of skin damage in the first submitted draft very briefly and given citation of sources for each of the mechanism skin damage (Section 4). However, the original draft version which I had prepared had detailed information about different mechanism of skin damage with almost the same list of citations.  

Based on the recommendation of one of the reviewer, I have incorporated more details about different mechanisms of skin damage in the first revision, almost like in the original draft, and as result most of the citations / references have not changed. However, one additional reference is included in Section 4 of the article (Reference 22).  Two new references (no 56 and 64) covering details of mechanism of action of different actives were also included in Section 6 of the revised draft. 

Reviewer 4 Report

The revision did not address my original comments, therefore recommend to reject.

Author Response

A figure giving a snapshot of various environment pollutants, visible effects of skin damage and formulation approaches to counter the effect of various environmental aggressors is included in the revised draft. (Figure 1)

A table giving guidelines for selection of actives depending on the type of skin damage and desired benefit is also incorporated in the revised draft. (Table1)   

The section 6 covers information about different anti-pollution actives and references 29-66 in the initial versions of the draft were webpages of supplier of the different actives. All the webpage references have been deleted and replaced by either literature published in different journal or by patents.

Round 3

Reviewer 4 Report

I am happy with the revision, suggest to publish as it is.